# H2-M and H2-O as Targeting Vehicles for the MHC Class II Processing Compartment Promote Antigen-Specific CD4+ T Cell Activation

**DOI:** 10.3390/vaccines9101053

**Published:** 2021-09-22

**Authors:** Lucia Lapazio, Monika Braun, Kaj Grandien

**Affiliations:** 1Sanofi-Aventis Deutschland GmbH, R&D, Biologics Research, Industriepark Hoechst, 65926 Frankfurt am Main, Germany; monika.braun79@gmx.de (M.B.); Kaj.Grandien@sanofi.com (K.G.); 2Medigene, Lochhamer Str. 11, Martinsried, 82152 Planegg, Germany

**Keywords:** vaccine, mRNA, antigen presentation, fusion protein, T cell activation

## Abstract

CD8 and CD4 T cell activation are both required for a strong and long-lasting T cell immune response. Endogenously expressed proteins are readily processed by the MHC class I antigen presentation pathway, enabling activation of CD8+ T cells. However, the MHC class II antigen presentation pathway, necessary for CD4+ T cell activation, is generally not sufficiently accessible to endogenously expressed proteins, limiting the efficiency of mRNA- or DNA-based vaccines. In the current study, we have evaluated the feasibility of using antigen sequences fused to sequences derived from the H2-M and H2-O proteins, two complexes known to participate in MHC class II antigen processing, for the enhancement of CD4 T-cell activation. We analyzed T cell activation after genetic immunization with mRNA-encoding fusion proteins with the model antigen ovalbumin and sequences derived from H2-M or H2-O. Our results show that H2-M- or H2-O-derived sequences robustly improve antigen-specific CD4 T-cell activation when fused to the antigen of interest and suggest that the approach could be used to improve the efficiency of mRNA- or DNA-based vaccines.

## 1. Introduction

An efficient vaccination strategy, leading to a strong and durable immunological memory, requires a specific and potent interaction between antigen-specific T cells and professional antigen-presenting cells (APCs), such as dendritic cells (DCs), and activation of both CD4+ and CD8+ T cells [1]. Traditionally, experimental vaccination campaigns using live-attenuated viruses for infectious disease indications or cancer immunotherapy have focused mainly on activating cytotoxic CD8+ T cells, using MHC class I-restricted antigens. In contrast to infectious (live) vaccines, subunit Ag are notoriously poor in eliciting protective CD8 T-cell responses, presumably because subunit antigens become insufficiently cross-presented by dendritic cells (DCs) and because the latter need to be activated to acquire competence for cross-priming. Similarly, in cancer immunotherapy with antigen-specific approaches using weakly antigenic targets such as tumor-associated antigens (TTAs), this strategy has elicited only a modest and transient cytolytic CD8+ immune response and only in a subset of immunized patients [2,3,4].

CD4+ T-cell responses play a key role in the help, maintenance, and promotion of memory CD8+ T cells, especially against weakly immunogenic antigens. Therefore, MHC class II antigen presentation and the subsequent activation of CD4 T cells are essential to obtain a strong and long-lasting immune response [1]. Efficient activation of CD4+ T cells may be a particular challenge for intracellular proteins which are, in general, poorly presented to CD4+ T cells, since they are not naturally directed to endosomal–lysosomal antigen-processing compartments to generate peptide–MHC class II complexes and activate naive CD4+ T cells [5].

In order to enhance naïve CD4+ T-cell activation, different approaches have been developed to more efficiently deliver antigenic epitopes to the MHC class II pathway or directly to the MHC class II antigen-processing compartment [1]. These approaches include coupling antigens to selected sequences of trafficking proteins known to reside in endo-lysosomal compartment pathways, such as invariant chain [6,7,8,9,10], LAMP-1 [11,12,13], DC-LAMP [11], and transferrin receptor [7], or to proteins which redistribute from lysosomes to MHC class II-containing late endosomal antigen-processing compartments upon activation of DCs, such as cathepsins or other acid proteases [14,15,16,17]. Yet another approach explores coupling antigen to the trans-membrane domain of MHC class I proteins [18].

In our study, we focused on proteins located in subcellular compartments associated with the MCH class II processing route, which has not been investigated for enhancing antigen presentation. We postulated that two enzymes, mouse H2-M and H2-O (and their human counterparts HLA-DM and HLA-DO), which are known to be involved in peptide loading of class II MHC molecules in endosomal compartments, are suitable candidates for improving antigen presentation [5]. H2-M and H2-O are two chaperone proteins, each consisting of an α and β chain, and which in a process called “editing” catalyze the removal of CLIP (a fragment of the invariant chain Ii) from the MHC II binding groove to replace it by an antigenic peptide [19]. The function of these non-classical MHC II proteins has been characterized in APCs, and H2-M is considered to play a key role in MHC II antigen presentation and CD4 T+ cell epitope selection [20,21,22,23,24,25]. The activity of H2-M is regulated by H2-O which in initial reports was described to suppress H2-M activity, but more recent data suggest a rather complex role of H2-O including fine-tuning of the epitope selection process [26,27]. Still, the precise role of this second MHC class-II-like protein remains to be fully elucidated [27]. In our studies, we tested both H2-M and H2-O as candidates for re-directing antigens to the MHC class II compartments through a set of mRNA constructs encoding for a fusion of these proteins with the model antigen ovalbumin. Our results indicate that H2-M and H2-O are suitable sorting-target signals for redirecting antigen to the MHC class II pathway, thereby enhancing the antigenic-specific CD4+ T-cell response.

## 2. Materials and Methods

### 2.1. Design of Fusion Protein Sequences

As a model antigen for evaluation of antigen presentation a fragment of chicken Ovalbumin was used, OVA139-386, which contains the MHC class I-specific H2-Kb-restricted epitope SIINFEKL (OVA257-264) as well as the epitope ISQAVHAAHAEINEAGR (OVA323-339) which binds to I-A(b) MHC class II. For the Invariant Chain (Ii) OVA fusion protein, a design similar to the one described by [7] was used. To the full-length Ii protein (NP_034675), the OVA139-386 sequence and a cMyc epitope tag (EQKLISEEDL) were fused to the C-terminus.

The H2-M and H2-O α and β proteins all share a similar structure consisting of a signal peptide, a luminal region, a transmembrane (TM) domain, and a cytoplasmic domain. For these proteins, two types of OVA fusion constructs were designed; a short form, where the luminal region was replaced with OVA139-386, and a long form where OVA139-386 was inserted between the luminal and the TM domains.

The H2-Mα-OVA short fusion protein contains amino acids 1–28 and 230–261 (from NP_ 0345516), respectively, separated by OVA139-386 and a Myc tag flanked by G4S linker sequences. The H2-Mα -OVA long fusion protein contains aa 1–231 and 230–261, respectively, separated by the same OVA-cMyc cassette. The H2-Mβ-OVA short fusion protein consists of aa 1–18 and 217–261 (from NP_034517), respectively, separated by the OVA-cMyc cassette. The H2-Mβ-OVA long fusion protein was generated from aa 1–218 and 217–261, separated by the OVA-cMyc cassette.

The H2-Oα-OVA short fusion protein was constructed from aa 1–27 and 216–250 (from NP_034518), respectively, again separated by the OVA-cMyc sequence cassette. The H2-Oα-OVA long fusion protein was generated from aa 1–217 and 216–250, separated by the OVA-cMyc cassette. The H2-Oβ-OVA short fusion protein was constructed from aa 1–28 and x-x (from NP_034519, respectively separated by the OVA-cMyc cassette. The H2-Oβ-OVA long fusion protein was generated from aa 1–221 and 220–271, separated by the OVA-cMyc cassette. The sequences for all H2-M and H2-O OVA fusion proteins are available in Appendix A.

An mRNA encoding only OVA139-386 plus a cMyc tag was used as a negative control. An mRNA encoding the unrelated protein Arah1 (AAB00861) was also used as a negative control.

### 2.2. mRNA In Vitro Transcription

cDNA sequences encoding the proteins of interest were generated by gene synthesis and cloned in the vector pmRNA (TriLink) which contains a T7 RNA polymerase promoter, a synthetic 5′ UTR, and a 3′ UTR derived from the mouse a-globin gene. Chemically modified (100% substitution with pseudouridine and 5-methyl cytidine), co-transcriptionally capped Cap1 mRNAs were synthesized by T7 RNA polymerase in vitro transcription (TriLink) as described previously [28].

### 2.3. Animals

C57BL/6 mice, Ly5.1 (B6.SJL-PtprcaPepcb/BoyCrl, CD45.1) mice, OT-I mice (C57BL/6-Tg(TcraTcrb)1100Mjb/Crl, CD45.2), and OT-II mice (C57BL/6-Tg(TcraTcrb)425Cbn/Crl, CD45.2) were purchased from Charles River Laboratory. Age-matched 6–12 weeks female mice were used for all the experiments. All immunizations were performed i.v. in the tail vein. All animal experiments were carried out in an AAALAC-certified animal facility in accordance with the German animal welfare law and approved by the local authorities.

### 2.4. Cells

Murine bone marrow-derived dendritic cells (BM-DCs) were generated by culturing bulk cells from bone cavities of C57BL/6 mice in RPMI medium (Gibco) supplemented with 1% Pen/Strep (Gibco), 50µM ß-mercaptoethanol (Gibco), 10% heat-inactivated FCS (Gibco) and 20 ng/mL of GM-CSF (Peprotech). After three days, fresh media supplemented with 20 ng/mL of GM-CSF was added to the culture. The cells were collected after six days of culture and used for the in vitro T cell proliferation assay studies. CD4+ T cells (OT-II cells) were isolated from splenocytes of OT-II mice using AutoMacs separation (CD4+ T Cell Isolation Kit, mouse, Milltenyi Biotec). Similarly, CD8+ T cells (OT-I cells) were obtained from splenocytes of OT-I mice using AutoMacs separation (CD8+ T Cell Isolation Kit, mouse, Milltenyi Biotec).

### 2.5. In Vitro T Cell Proliferation Assay

To assess the CD4+ T cell activation in vitro, electroporation was selected as the best technique to deliver mRNA into the cytoplasm of BM-DCs. DCs (2 × 10^6^) were electroporated (1 pulse, 325 V, 150 µF) in 50 µL OptiMEM with a total of 1 µg of IVT mRNA using a GenePulser Xcell (Bio-Rad). CD4+ OT-II or CD8+ OT-I T cells, respectively, were isolated as described above and labeled with the CellTrace dye (Invitrogen) to assess the CD4+ and CD8+ T cell proliferation status by flow cytometry. Successful CellTrace staining was routinely controlled before injection of the labeled donor cells into the recipient animals. The co-cultivation was performed for 3 days at 37 °C at a 4:1 ratio of T cells per DC.

### 2.6. In Vivo T Cell Proliferation Model

For the assessment of the antigen-specific CD4+ T cell or CD8+ T cell activation and expansion, an adoptive mouse model was used. CD4+ T cells or CD8+ T cells were isolated as described above. Approximately 1.5 × 10^6^ cells were labeled with CellTrace dye (Invitrogen) and adoptively transferred at day 0 by i.v. injection to recipient Ly5.1 mice (B6.SJL-PtprcaPepcb/BoyCrl). The following day, the animals were immunized i.v. with a total og 5 µg of mRNA formulated with RNAiMAX (Life Technologies, Darmstadt, Germany) using a ratio of 2:1 of µL RNAiMax: µg mRNA in OptiMem (Gibco, Darmstadt, Germany). This mRNA formulation protocol has been described previously for Lipofectamine RNAiMax [29], leading to an almost exclusive mRNA expression in splenic DCs. Spleens were collected at day 5 and CD4+ T cells or CD8+ T cells, respectively, were isolated using the CD4+ or CD8+ T Cell Isolation Kits (mouse, Milltenyi Biotec, Bergisch Gladbach, Germany) and analyzed by flow cytometry.

### 2.7. Antibodies and Flow Cytometry

The following fluorochrome-conjugated antibodies (mAb) were used. For the in vitro assay, Dead Cell Marker-FarRed (Life Technologies), CD3-DAPI (Biolegend, San Diego, CA, USA), CD45-BV510 (Biolegend), CD4-FITC (eBiosciences, Thermo Fisher Scientific, Waltham, MA, USA), CD8-PE (eBiosciences), CD11c-PE-Cy7 (eBiosciences), and CellTrace Far Red-APC (Invitrogen, Waltham, MA, USA) were used. For the in vivo studies, Dead Cell Marker-Aqua (Life Technologies), CD3-DAPI (Biolegend), CD45.2-PE (eBiosciences), CD45.1-eFluor780 (eBiosciences), CD4-FITC (eBiosciences), CD8-BV421 (BD), CD11c-PE-Cy7 (eBiosciences), CD25-PE Cy7 (eBiosciences), CD19-BV711 (BD) and CellTrace Far Red-APC (Invitrogen) were used. For cell surface analysis of proliferation, CD4+ T cells were collected after the co-culture, washed two times with FACS Stain Buffer (BD), and stained with the indicated antibody mastermix for 20 min at 4 °C. Then, the cells were harvested and fixed with Fix/Perm (eBioscience) and finally analyzed for Celltrace dilution with flow cytometry. Samples were acquired on a BD LSRFortessa flow cytometer (BD Biosciences, Franklin Lakes, NJ, USA) and data were analyzed using FlowJo v10 (Tree Star Inc., San Carlos, CA, USA). Calculations of percent proliferated T cells were essentially performed according to [30].

### 2.8. Statistical Analysis

Statistical analysis was performed using one-way ANOVA parametric test, GraphPad Prism. *p* < 0.01 was considered significant. Post hoc analysis using the Dunnett method was applied when possible.

## 3. Results

### 3.1. Design and Characterization of mRNAs

We speculated that proteins co-localized to the cellular compartments involved in antigen presentation could be utilized to enhance T cell activation through the MHC class II complex. The H2-M and H2-O proteins, mouse homologs of the human HLA-M and HLA-O proteins, were identified from the literature as strong candidates since these proteins are known to be involved in the MHC class II antigen presentation route. Both proteins are complexes of an alpha and a beta chain and we decided to generate antigen fusion protein for all four proteins H2-Mα, H2-Mβ, H2-Oα, and H2-Oβ. As a model antigen, we selected ovalbumin (OVA) and the identified protein sequences were used to design protein fusion sequences with OVA (Figure 1 and Table 1, see Appendix A for full sequences).

We designed two categories of fusion proteins for the H2-M and H2-O proteins, denoted as long and short forms, in order to increase the probability of identifying a suitable fusion protein design. The H2-M and H2-O proteins share a similar protein domain structure; a signal peptide (SP) precedes a luminal domain, which is followed by transmembrane (TM) and cytosolic domain (CD). Long fusion proteins were designed by inserting the OVA coding sequence between the luminal domain and the transmembrane domain and retaining all domains of the original H2 protein. For the short fusion proteins, the luminal domain was fully replaced with the OVA coding sequence.

Subsequently, mRNAs corresponding to all the designed fusion protein sequences were generated by in vitro transcription as described previously [28].

### 3.2. Enhanced CD4 T-Cell Activation In Vitro Mediated by H2-M and H2-O OVA Fusion mRNAs

To analyze the effect of the mRNA-encoded fusion proteins constructs on CD4 T-cell activation, several independent experiments were performed using a murine CD4+ T cell in vitro proliferation assay. Briefly, OVA-specific CD4+ T cells were obtained from spleens of OTII transgenic mice, a mouse strain with highly elevated numbers of OVA-specific CD4 T cells [31]. The cells were labeled with CellTrace dye (Invitrogen) and co-cultured for 3 days with mRNA-transfected BM-DCs. The activation of T cells results in cell proliferation and consequently CellTrace dye dilution, thus leading to a decrease in median fluorescence intensity (MFI) as a measure of T cell proliferation. An mRNA encoding the OVA construct without any protein fusion and mRNA encoding non-related peanut antigen Arah1 were used as controls. The compiled results of three independent experiments are shown in Figure 2.

We observed that the OVA encoding mRNA without fusion protein sequences (OVA-mRNA) resulted in values close to the Arah1-mRNA, indicating that no significant specific CD4+ T-cell proliferation was elicited, which is in agreement with previously published observations of inefficient activation of CD4 T cells with antigen-encoding mRNAs without fusion protein sequences [7,18]. In contrast, all constructs, except for long forms of H2-M β and H2-O α, generated a significant decrease in the CellTrace MFI, indicating significant CD4 T-cell proliferation (Figure 2). In addition, combinations of the mRNAs encoding the respective α and β chains (here referred to as “combos”) also generated a robust CD4 activation.

### 3.3. H2-M and H2-O OVA Fusion mRNAs Increase the CD4 T-Cell Activation and Expansion in a Murine In Vivo Model

To confirm the in vitro murine data and to determine the impact of the selected fusion protein candidates in vivo, a suitable murine in vivo model was used. Briefly, OVA-specific CD4+ T cells were isolated from CD45.2 OTII transgenic mice, labeled with CellTrace dye (Invitrogen), and adoptively transferred i.v. into CD45.1 Ly5.1 host mice. The following day, OVA-encoding mRNA formulated with Lipofectamine RNAiMAX [28] was administered i.v. to the host mice. Five days after the immunization, mice were sacrificed, splenocytes were isolated, and the CD4+ T cells were analyzed for activation and proliferation by flow cytometry (Figure 3A). As a positive control, we utilized an mRNA encoding for Invariant Chain (Ii) fused to OVA—a construct design previously described [7]—and mRNA encoding for OVA only as a negative control.

For the in vivo studies, we decided to focus on the short forms of H2-M and H2-O fusion proteins since all short forms, but not all long forms, had demonstrated a robust activation in vitro. To assess T-cell proliferation, we quantified the decline of the vital CellTrace dye signal in proliferating CD4+ T cells. We determined the enhancement in donor CD4+ T-cell proliferation by measuring either the percentage of all proliferating donor cells (Figure 3B) relative to the total number of donor cells, or the percentage of cells in the last division peak (fully proliferated cells, Figure 3C). We observed an enhanced proliferation of OVA-specific CD4+ T cells for mice injected with mRNAs encoding H2-Mα, H2-Oα, and β and the respective combos compared to the non-fusion control mRNA (OVA-mRNA). H2-Mβ reached only about 30% of enhanced proliferation instead of 90% reached by the H2-Mα mRNA and the H2-O mRNAs (Figure 3B). Subsequently, we also looked in more detail into the proliferation data to find out if there are any differences between the mRNAs in terms of the state of the proliferation. Looking at the last division peak, we found a lower frequency of CD4+ T cells in the case of mice immunized with H2-Mα in comparison with the frequency of the other fusion mRNAs, except for H2-Mβ mRNA which did not show an increase in proliferation (Figure 3C).

Next, we investigated whether the enhancement in CD4+ T-cell proliferation due to the presence of the fusion proteins was also correlated to an expansion of the donor CD4+ T-cell population. The T cell expansion was calculated as the percentage of donor CD4+ T cells relative to the total number of CD4 T+ cells (Figure 3D). We observed that in terms of CD4+ T cell expansion the H2-Mβ fusion protein did not show an increase in donor CD4+ T cell numbers, indeed the results are comparable to the control OVA-mRNA. In contrast, we found that the H2-Mα, H2-Oα and -Oβ and the α/β combos generated an increase in donor CD4+ T cell expansion (Figure 3D). More specifically, we observed that five days post injection of the mRNA encoding OVA antigen fused to H2-Mα, the CD4+ donor T cell expansion was enhanced up to 3.5% while the OVA-mRNA showed a maximum of 0.5%. A similar increase in CD4+ T cells was also observed for the mRNAs encoding H2-Oα and -Oβ and the α/β combos. This result is in accordance with the CD4+ T cell proliferation data.

Finally, we investigated if the presence of fusion protein sequences had an impact on the proliferation of CD8+ T cells. For this, we utilized CellTrace-labeled CD8+ T cells isolated from OTI mice to generate Chimeric Ly5.1 animals. As shown in Figure 4, all analyzed fusion proteins induced CD8 T+ cell proliferation as well as OVA without fusion protein sequences.

## 4. Discussion

Our study demonstrates for the first time that the coupling of antigen-derived protein sequences to H2-M or H2-O protein sequences promotes the activation and proliferation of CD4 T cells. More specifically, we show that in vitro or in vivo transfection of DCs with in vitro transcribed mRNAs coding for fusion protein sequences between antigen and H2-M or H2-O leads to potent activation of CD4+ T cells and of CD8+ T cells.

One limitation of mRNA-encoded antigens intended for immune stimulation is that by default such antigens will reside in the cytoplasm and thus are poorly accessible for the MHC class II antigen presentation pathway, as this pathway is not well reached by cytoplasmic proteins. Thus, these proteins will mainly be presented in the context of MHC class I molecules [32]. Today, it is well-accepted that not only CD8+ T-cell activation but also CD4+ T cell activation is required to obtain a potent and long-lasting immune response, and methods improving CD4+ T cell activation are an important enhancement of mRNA-based immunotherapies [1].

The growing knowledge of the MHC class II presentation pathway has given input for rational approaches to improve CD4+ T-cell activation [5]. Genetic modifications to antigen-coding sequences have been explored previously by other groups, aiming to enhance Ag-specific CD4+ T cell activation, [6,7,8,9,10,11,12,13,14,15,16,17,18]. Our studies demonstrate for the first time that the H2-O and H2-M proteins are well suited for enhancing MCH II antigen presentation, adding to the repertoire of known antigen targeting sequences [6,7,8,9,10,11,12,13,14,15,16,17,18] which may be explored for optimizing antigen presentation and CD4+ T cell activation.

H2-M and H2-O (and their human counterparts HLA-DM and HLA-DO) are two chaperone proteins, each consisting of an α and a β chain, which are localized in the antigen-processing compartments [26]. Both proteins are involved in the replacement of CLIP (a protein fragment derived from Invariant Chain), located in the peptide-binding groove of the MHCII complex, with peptides derived from exogenous antigens. H2-M acts as an enzyme to catalyze peptide exchange, while the mechanism of action of H2-O has been under debate. The current dogma is that H2-O inhibits H2-M, but recent findings suggest a new active role of this protein and this new model suggests that H2-O fine-tunes the epitope selection process and might be important for sorting of self vs. non-self peptides [26].

Given the strong co-localization and functional interaction between MHC II and H2-M and H2-O, respectively, we speculated that coupling parts of H2-M and H2-O to an antigen peptide would change its route and relocate it into the compartments where the antigen-derived peptides are loaded on the MHC class II complex. As described above, we generated two types of fusion proteins, the long forms retaining the entire sequence with the antigen sequence inserted before the transmembrane domain, and the short forms with the luminal domain removed.

Indeed, in our study, we found that mRNAs encoding H2-M and H2-O antigen fusion proteins generated an increase in CD4+ T cell activation and proliferation. First, the in vitro results demonstrated that the H2-M and H2-O short forms are very potent in the stimulation of OVA-specific CD4+ T cell proliferation—this is also true for the long forms of H2-Mα and H2-Oβ as well as for the combos. However, the long forms of H2-Mβ and H2-Oα did not perform better than the control. Possibly, the large forms are more susceptible to incorrect folding or inefficient expression, which might be improved when these proteins are co-expressed as combos.

Based on our in vitro observations, we selected H2-M and H2-O short forms as top leads to be tested in a murine in vivo model. We found a substantial increase in terms of CD4+ T cell proliferation and expansion when mice were injected with mRNA encoding H2-Mα, H2-Oα, and β and the respective α/β combos. However, H2-M β mRNA did not increase the percentage of CD4+ donor T cells in vivo. It should be noted that H2-M β was less potent than the other short forms in vitro, which might translate to a complete loss of efficiency in vivo.

When the proliferation status of the CD4+ donor T cells was analyzed in more detail, we found that more than 30% of CD4+ donor T cells were in the most proliferated state for the H2-Mβ mRNA and for both H2-O-mRNAs as well as the combos. The best performing fusion-mRNA was H2-O β, which increased the proliferation of OVA-specific CD4+ T cells to more than 80%.

Importantly, we noticed that using H2-M and H2-O fusion antigens CD8+ T cell activation via MHC class I presentation was not affected or sometimes even improved. Thus, the use of the H2-M and H2-O fusion protein described here will improve CD4+ T cell activation and at the same time could trigger a potent CD8+ T cell activation. In our studies, we used the well-established model antigen OVA because of the existing tools for CD4 and CD8 epitopes for this antigen and the knowledge of the MHC class I and class II specific epitopes for following, in a sensitive manner, the CD4 and CD8 responses. Future studies evaluating other antigen candidates fused to H2-M and H2-O sequences in an mRNA vaccine approach, also measuring the subsequent effector functions such as antibody secretion, could be valuable to further validate the general use of H2-M and H2-O for improving genetic vaccines. From our results on improved CD4 T cell activation, we are not able to draw any precise conclusions about the mechanisms behind this process. Additional studies using specific inhibitors of proteins involved in the MCHII antigen presentation pathway might be able to shed further light on the mechanistic details.

Collectively, the in vitro and in vivo mouse data presented here do establish that H2-Mα and H2-Oα and β are novel tools to increase antigen presentation via the MHC class II pathway when introduced into the cells as an mRNA vector. Furthermore, the chimeric antigen protein fusion designs proposed in this work can be used as tools for improving genetic vaccines by improving CD4+ T cell activation. We can foresee the application of these fusion proteins in different types of mRNA-based immunotherapies to enhance an otherwise poorly activated CD4+ T cell response.

## Figures and Tables

**Figure 1 vaccines-09-01053-f001:**
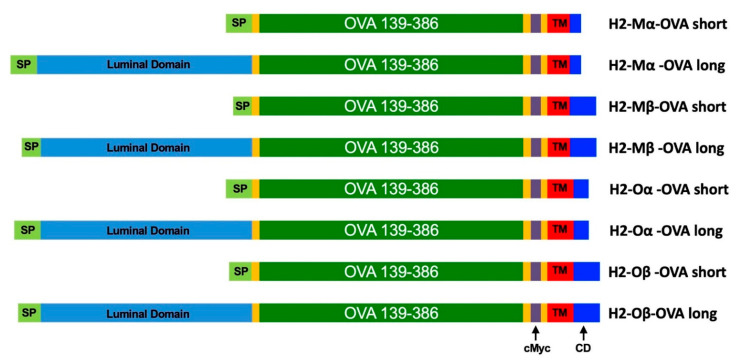
Schematic structure of mRNA-encoded H2-M OVA fusion protein constructs included in the study. Abbreviations: SP: Signal peptide (bright green), cMyc: cMyc epitope tag, TM: Transmembrane domain (red), CD: Cytoplasmic domain (dark blue). Yellow: Linker (GGGGS) sequences.

**Figure 2 vaccines-09-01053-f002:**
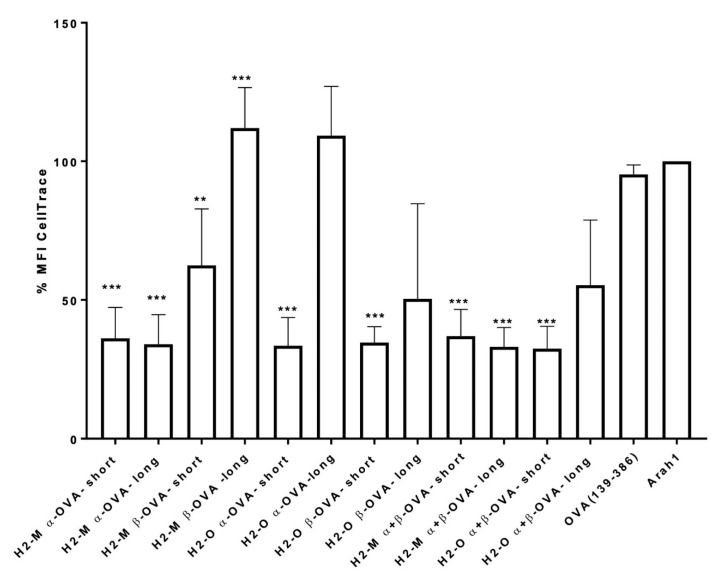
Improved CD4+ T cell activation in vitro mediated by H2-M and H2-O sequences. Fusion of OVA to H2-M or H2-O results in increased OVA-specific CD4+ T cell proliferation in comparison with unrelated mRNA (Arah) or non-fusion mRNA (OVA136-386). To assess the proliferation, murine OVA-specific CD4+ T cells were co-incubated for 3 days together with BM-DCs transfected with OVA-mRNA. The graph shows results from three independent experiments. Data shown are MFI (median fluorescence intensity) values of the cell membrane dye CellTrace normalized to the signal of the negative control Arah1. Statistics performed using one-way ANOVA. Control column Arah1. *** *p* < 0.001, ** *p* < 0.002.

**Figure 3 vaccines-09-01053-f003:**
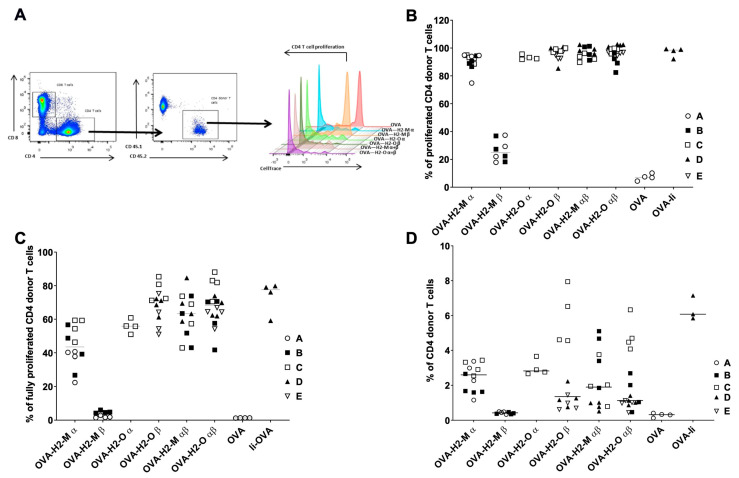
Enhanced CD4+ T cell proliferation in vivo mediated by the insertion of H2-M and H2-O fusion-protein sequences. (**A**) Quantification of in vivo CD4+ T cell proliferation generated by coupling OVA to selected trafficking signal sequences. Flow cytometry plots and histograms showing the analysis and quantification of the decline in vital cell tracer dye staining in proliferating CD4+ T cells. CD45.1 mice were adoptively transferred i.v. with CellTrace-labeled CD45.2/CD4/OTII cells and 24 h later immunized i.v. with formulated OVA-fusion mRNAs formulated with Lipofectamine MessengerMax. T-cell proliferation was measured at 5 days post CD4+ T cell transfer by flow cytometry after staining with the indicated fluorescent antibodies. CD4+ T cell proliferation was determined by measuring the percentage of all proliferating cells and the percentage of cells in the last division peak, respectively, as shown in the graphs. (**B**) In vivo quantification of CD4+ T-cell proliferation in all stages generated by coupling OVA to H2-M and H2-O sequences. Enhanced CD4+ T-cell proliferation mediated by H2-M and H2-O sequences. Data are shown as frequency of proliferating donor CD4+ T cells 3 days after the injection of 5 µg of mRNA. T-cell proliferation was determined by measuring the percentage of all proliferating cells. An mRNA-encoding part of the sequence of Invariant chain (Ii) was used as a positive benchmark. Data shown are a combination of five independent experiments (A–E), four female mice per group. (**C**) In vivo quantification of CD4+ T-cell proliferation in the latest stage generated by coupling OVA to H2-M and H2-O sequences. Data represent the percentage of proliferating CD4+ donor T cells in the highest proliferation stage, determined by measuring the percentage of proliferating cells in the last division peak. (**D**) Quantification of donor CD4+ T cell expansion generated by coupling OVA to H2-M and H2-O sequences. Data represent the percentage of donor CD4+ T cells relative to the recipient CD4+ T cells. Five independent experiments are represented (A to E), four mice per group. Invariant chain fusion protein (Ii) was used as a positive benchmark.

**Figure 4 vaccines-09-01053-f004:**
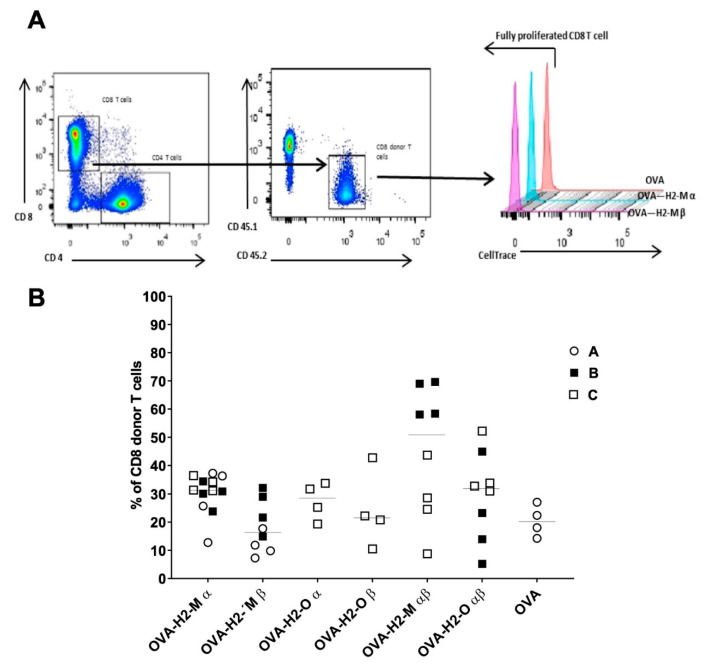
CD8+ T cell proliferation mediated by the insertion of H2-M and H2-O trafficking signal sequences. (**A**) In vivo quantification of CD8+ T cell proliferation generated by coupling OVA to selected trafficking signal sequences. Flow cytometry plots and histograms showing the analysis and quantification of the vital cell tracer dye decline staining in proliferating CD8+ T cells. CD45.1 mice were adoptively transferred i.v. with mouse CellTrace labeled CD45.2/CD8/OTII cells and 24 h later immunized i.v. with 5 µg of formulated OVA-fusion mRNAs. T cell proliferation was measured at 5 days post-DC transfer by flow cytometry after the staining. T cell proliferation was determined by measuring the percentage of CD8 proliferating T cells, as shown in the graph. All groups showed were fully proliferated. (**B**) In vivo CD8+ T cell expansion generated by coupling OVA to H2-M and H2-O sequences. Data in the graph represent the CD8 T cell expansion generated by H2-M and H2-O sequences. Three independent experiments are shown (A–C), four female mice per group.

**Table 1 vaccines-09-01053-t001:** Fusion proteins used for murine T cell activation assays. Overview of the protein sequences selected for the study.

Fusion Protein	ORF Length	AA Length	mRNA Size	Reference Sequences	Amino Acids
H2-M alpha chain-OVA (short)	1002	333	1.278	NP_ 0345516	1–28; 230–261
H2-M alpha chain-OVA (long)	1611	536	1887		1–231; 230–261
H2-M beta chain-OVA (short)	1017	338	1.293	NP_034517	1–18; 217–261
H2-M beta chain-OVA (long)	1611	536	1.887		1–218; 217–261
H2-O alpha chain-OVA (short)	1008	335	1284	NP_034518	1–27; 216–250
H2-O alpha chain-OVA (long)	1578	525	1.854		1–217; 216–250
H2-O beta chain-OVA (short)	1065	354	1.284	NP_034519	1–28; 219–271
H2-O beta chain-OVA (long)	1644	547	1.920		1–221; 219–271
Ii-OVA	1434	477	1.710	NP_034675	1–214

## Data Availability

The authors confirm that the data supporting the findings of this study are available within the article and its Appendix A. Raw data sets are available from the corresponding author, L.L., upon request.

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
