# Peer review of "H2-M and H2-O as Targeting Vehicles for the MHC Class II Processing Compartment Promote Antigen-Specific CD4+ T Cell Activation"

_vaccines, 2021, doi:10.3390/vaccines9101053_

Round 1
Reviewer 1 Report
The topic of the manuscript is interesting nd fits well the scope of the journal. The reviewer feels it can be accepted after some minor amendments.
(1) Please double check the writing. Although English is not the fist language of the reviewer, he notice various typos / grammar mistakes.
(2) Why use one way ANOVA? Had the authors examined the normality of their data? Also, if the variables are related, t-test or its non-parametric equivalent may be more appropriate.
(3) Mice are used in in vivo model. Is there any known significant species difference between human and mouse?
Reviewer 2 Report
Activation of CD4 T cells is an important goal in successful vaccine development and Lapazio et al describe a new method to target antigens to the MHC class II antigen presentation pathway. In essence, they generate fusion proteins containing the antigen of interest and target these antigens to MHC class II molecules by fusing them to the transmembrane and cytotosolic domains of MHC class II chaperone molecules. The rationale is the fusion proteins will likely be degraded into peptides in the same region of the cell where MHC class II peptide editing occurs, ensuring efficient antigen presentation. Overall the data presented suggests such a strategy works, however, I have several concerns that preclude publication in Vaccines at this time.
- The authors present cell proliferation data in a curious way, either as % reduction in MFI (Figure 2), or % of proliferated (fully or otherwise) T cells (Figure 3). A more appropriate way to report this data is either the division index or the proliferation index. The authors should re-analyze the data sets and present these values which are more acceptable to the field.
- A central premise to the study is that these proteins are somehow targeted to MHC class II by the presence of H2-M or H2-O components. However, in the "short" forms of these constructs, only the transmembrane and cytosolic domains are present. Are these constructs presented because they are simply targeted to endosomal compartments? Would other endosomal targeting sequences (apart from Ii, which is discussed) achieve similar results?
- It is interesting that the long form of H-2MB and H-2OA do not activate T cells. Does the presence of the luminal domain prevent the protein from localizing to the correct regions of the cells?
- A good deal of background information was omitted from the introduction tat should at least be briefly discussed. Several recent studies have suggested roles new roles for H-2O (Sadegh-Nasseri and colleagues have done a great deal of this work) and this should be discussed rather than leaving the reader with the impression that the functions of H-2O are unknown as suggested by the referenced review from 2006. Similarly, a discussion of known cases of endogenous MHC class II presentation should be discussed. Papers from the Long group, Ostrand-Rosenberg group, and the Eisenlohr group to name but a few, have suggested that endogenous antigens can be presented via MHC class II. Several excellent review articles have been written on the subject and I would encourage the authors to reference these in the introduction.
- Minor point. Line 80, while the peptide is the same, the authors should refer to it as I-Ab restricted and not I-Ad as they are using B6 mice which express I-Ab.
Reviewer 3 Report
The authors present a very well written and well constructed study that describes an approach for enhancing presentation of antigens via MHC II to transgenic OTII T-cells. Identifying strategies for targeting protein antigens to the MHC II pathway is critical for improving therapeutic and vaccine strategies that elicit CD4+ T-cells responses. Although the authors present data that demonstrates that their H-2M and H2-O/ovalbumin constructs can successfully activate OTII T-cells they fall short in demonstrating that the antigens presented on BM-DC are specifically processed by the MHC II processing pathway. Further experiments to demonstrate this will be required.
Major points
- The authors clearly demonstrate that their H-2M and H-2O/ovalbumin constructs are capable of presenting antigens to OTII T-cells, however it is less clear whether this is through targeting to the MHC II antigen processing pathway since antigen is also presented through MHC I. Further experiments to inhibit the MHC II antigen processing pathway using specific inhibition or knockdown of proteases would be required to make this conclusion. Further, inhibition of the MHC I pathway would demonstrate specificity. Alternatively, demonstration that the products of this construct colocalize with defined MHC II processing endosomes/lysosomes would support the hypothesis and title.
- All of the constructs integrate the same ovalbumin sequence for this model antigen which leaves open the possibility that the use of this sequence alone could explain the targeting of these constructs to the MHC II processing pathway. Can the authors demonstrate elicitation of a response to an antigen other than ovalbumin? This will be critical for demonstrating that alternative antigens can be targeted to the MHC II presentation pathway and strengthen the potential for translation.
- For the in vivo activation experiments (Figure 3 and 4) the authors chose to detect proliferation at 5 days post challenge. Most or all of the cells had completely diluted the cell-tracer at this point and were detected in a single peak, suggesting that they had terminally differentiated or lost the marker dye. In the absence of multiple peaks of division, analysis of the OTII cells for the upregulation of activation markers would be helpful for determining whether the OTII T-cells had been specifically activated. Alternatively this should be repeated at an earlier timepoint in a range that is able to detect multiple division peaks.
Minor points
- Figure 3A and Figure 4A are unreadable. These should be submitted at a higher resolution. These are difficult to interpret as they are in the proof.
- In the statistics section specify the pairwise comparison test used after the One-way ANOVA.
Round 2
Reviewer 2 Report
The authors have addressed the concerns.
Author Response
We are grateful to the Reviewer 2 for his/her constructive input.
Reviewer 3 Report
The authors submit a revised version of their manuscript that aims to address prior critiques. There is not a sufficient improvement of this manuscript to warrant publication in Vaccines.
Major Points.
- Yes OTII T-cells exclusively recognize antigens presented by MHC II, however the authors are extending their conclusion to say that their targeting approach directs antigens to the MHC II processing pathway. This conclusion warrants a higher level of rigor which will include inhibition or knockdown of proteases/enzymes that actually process MHC II antigens through the lysosomal/endosomal pathways. The authors provide no evidence that these targeted constructs actually get to the classical MHC II processing pathway.
- Based on the reply to point number 2 from the previous critique it is evident that this system may only be observed using the OTII T-cell system. This decreases my enthusiasm for the manuscript since there is not evidence that this approach will induce an immune response in animals with a normal repertoire of T-cells. In the absence of this data, the finding could just be an artifact of the OTII system.
- In regards to the reply to previous critique 3, the authors are correct that the Cell Trace Dye is covalently linked in the cytosol of the cells and that the signal is diluted however this reviewer has a difficult time differentiating low Cell Trace from negative since that control is not included.
- In regard to minor point 2, One-way ANOVA is the appropriate test however a post hoc test like Tukey, Newman-Keuls, Scheffee, Bonferroni or Dunnett should have been used.
Round 3
Reviewer 3 Report
The authors submit a revised version of their previously submitted manuscript that uses H2-M and H2-O sequences in a vaccine approach to promote MHC II OVA specific T-cell responses. The authors provide compelling arguments for publication of this manuscript in Vaccines.
Minor criticism
- The authors state that there is an "improvement" or "enhancement" of the T-cell response when using H2-M or H-2O sequences as targeting vehicles to the MHC II pathway. However, there are minimal comparisons to alternative targeting strategies and the comparisons to unrelated mRNAs or to non-fusion mRNA present a very low bar. The only relevant fusion control is OVA fused to invariant chain which demonstrates equivalent proliferation/activation. I think it would be fair to say that these constructs "promote" MHC II activation, however its not clear if this is an improvement over other published strategies.
- Replace "improve" in title to "promote"
